# Development of Novel Forms of Fungal Art Using *Aspergillus nidulans*

**DOI:** 10.3390/jof7121018

**Published:** 2021-11-28

**Authors:** Ofer Grunwald, Ety Harish, Nir Osherov

**Affiliations:** Department of Clinical Microbiology and Immunology, Sackler School of Medicine, Tel-Aviv University, Tel-Aviv 69978, Israel; ofer@tirgumation.com (O.G.); etyharish@mail.tau.ac.il (E.H.)

**Keywords:** fungal art, *Aspergillus nidulans*

## Abstract

Fungi are embedded in human culture, tradition, and art, and have featured as inspirational and visual motifs. Psychedelic and medicinal mushrooms have been sculpted, painted, and ingested by our ancestors since prehistory. In modern times, the growing divide between the arts and sciences has delegated fungal art to a niche activity, with the bulk of the focus being on mycelium as a biomaterial. A collaboration between a multidisciplinary artist and a research laboratory, specializing in the molecular study of Aspergillus molds, has allowed us to develop new forms of mycelial art. We describe in detail the development of fungal art techniques using nutrient-rich agar containing *Aspergillus nidulans* conidia spotted on glass acrylic surfaces or impregnated onto etched acrylic blocks. This approach generates visually and temporally dynamic artwork that is user-friendly, safe, relatively resistant to contamination and easily scalable. Moreover, it offers countless avenues of artistic development based on the diversity of colors, textures and shapes afforded by different fungal species.

## 1. Introduction

The scientific study of fungi (mycology) has traditionally been separated from the world of fungal art with almost no cross-talk or interaction between scientists and artists, despite the fact that up to the 18th century, science and art were studied interchangeably by scholars and polymaths. The separation between the sciences and arts deepened as the body of scientific knowledge grew exponentially and became increasingly specialized. The lack of cross-fertilization between the disciplines is detrimental to the development of both fields, especially in the current era, where multidisciplinary approaches are necessary to understand complex phenomena [1,2,3].

There are remarkably few scientific publications in the field of fungal art, leaving the creations of designers and artists almost entirely under the radar of scientists. Most designers have focused on fungal mycelium as a structural material which is used as physical support for their work. Extensive research has also been carried out in developing mycelium-based materials such as meat substitutes, artificial leather, textiles for clothing, synthetic wood-like composites, or packing material. Others have taken advantage of the role of fungi as decomposers to develop bio-refining and bioremediation applications [4,5]. Artists have created numerous carvings of mushrooms, including the Pilzstein stone mushroom from El Salvador (ca 300 BC-250 AD), apparently related to the consumption of psychedelic mushrooms, and many other contemporary works which recreate fungi using varied sculpting materials such as ceramics, textiles, yarn and wire. More recently, several artists have incorporated fungal fruiting bodies as visual elements in sculptural compositions [2,6,7,8,9,10,11,12]. Paintings and photographs of fungi are also a small but thriving niche art form [13,14].

In their yearly agar art contest, the American Society of Microbiology (ASM) has recently reignited interest in using actual living microorganisms as a painter’s pallet [15]. Paintings are completed by applying different microorganisms on agar-containing Petri dishes, which are incubated until the desired picture composed of growing colonies is achieved. Surprisingly, most of this artwork is made using bacteria even though fungi offer a huge and dynamic diversity of colors, textures and shapes.

Here, the Osherov Lab, which specializes in the molecular study of the pathogenic mold *Aspergillus fumigatus* [16], has collaborated with the multidisciplinary artist, Ofer Grunwald [17], to develop two novel fungal art forms using live mycelium of *Aspergillus nidulans*. The first, inspired by an Aboriginal dot painting [18], employs droplets of agar medium containing viable *A. nidulans* conidia to form abstract patterns on black acrylic glass. The second develops this concept further and uses computer numerical control (CNC) lathing to etch channels in acrylic glass, which are then inoculated with the *A. nidulans* conidial agar to form an image that subsequently evolves as the fungal colony develops. We believe that the visual impact of the resulting artworks, the unique physical and conceptual attributes of fungal mycelium, coupled with the robustness of the developed technique and its scalability into large-scale artworks, opens the door to the tremendous untapped potential of mycelium as a novel painting medium.

## 2. Materials and Methods

### 2.1. Media and Strains

*A. nidulans* strain R153 was grown for 48 h on YAG agar plates which contains: 0.5% yeast extract, 1% glucose, 10 mM MgSO_4_, vitamin mix, trace elements and 1.5% agar (*w*/*v*) at 37 °C. After incubation at 37 °C for 48 h, conidia were harvested in 0.02% Tween 20. Conidia were counted using a hemocytometer.

### 2.2. Dot Painting

*A*. *nidulans* was grown on YAG agar for 48 h and the conidia were harvested and counted. YAG agar medium was prepared and autoclaved in a volume of 250 mL in 500 mL bottles, then allowed to cool to 50–55 °C while being stirred on a hot plate. The temperature was continuously monitored with a digital thermometer. Conidia were added to the stirring YAG agar to a final concentration of 10^3^–10^7^ conidia/mL and used within 1 h. In total, 30–50 µL volumes of conidial YAG agar were rapidly spotted on the plastic surface of the Petri dishes or black acrylic glass, which were then placed in a humid chamber and incubated at 25–37 °C for 24–72 h. Humid chambers for the large black acrylic glass artworks were custom built from acrylic glass. Humidity was maintained with wet paper towels lining the chamber.

### 2.3. Computer Numerical Control Lathing and Etching Prints

Historical aerial photographs were used as reference images and manually traced into a vector-file format on a computer. Each file was then exported to a CNC lathe machine and cut as 3 mm-deep channels into 20 mm-thick blocks of transparent acrylic glass. YAG agar infused with *A. nidulans* conidia at a concentration of 10^7^ conidia/mL was prepared using the above method. The inoculated YAG agar was then injected into the lathed channels using various pipettes. The blocks were covered with 8 mm-thick acrylic glass lids and the sides were sealed with two layers of adhesive medical tape. The prints were then left to develop at room temperature indefinitely.

## 3. Results

### 3.1. Calibration and Setup of the Fungal Dot Painting Technique

We chose to use *A. nidulans* for this work because it is a well-characterized model organism and is generally regarded as safe (GRAS). It produces large quantities of asexual spores (conidia) on aerial structures called conidiophores, and laboratory strains with different conidial pigmentation and mycelial textures have been generated, producing a broad palette for creating varied art forms [19]. *A. nidulans* conidia are relatively tightly attached to the conidiophores, reducing the risk of contamination and cross-spread during work. We used *A. nidulans* strain R153 (*wA2:pyroA4*) since its growth depends on supplementation with pyridoxine (vitamin B6), limiting its growth to supplemented media, and because it produces non-melanized conidia, confining our painter’s palette to monochromatic hues ranging from white-cream to pale brown (Figure 1A). This limited, monochromatic palette was an aesthetic and conceptual requirement for the intended artworks. *A. nidulans* strain R153 was grown for 48 h at 37 °C on YAG agar plates. Conidia were collected, counted and resuspended at a concentration of 10^5^ conidia/mL in liquid YAG agar kept at a temperature of 50–55 °C. Conidial viability was maintained by using the suspension within 1 h of preparation. The conidial agar suspension was inoculated on the surface of an empty plastic Petri dish at different volumes (10–50 µL YAG agar) and incubated for up to 48 h at 37 °C. Results show consistent fungal growth and clearly defined mycelial dots forming after 24 h at droplet volumes of 20 µL and larger, reaching a stable composition after 48 h (Figure 1B). We found that fungal growth was strictly limited to within the agar droplet, with negligible contamination occurring. Inoculation of conidial agar suspensions varying in both volume (10–50 µL) and concentration (10^3^–10^5^ conidia/mL) showed that diluting conidia from 10^5^ to 10^4^ to 10^3^ conidia/mL delayed the appearance of mycelial dots from 24 h to 48 h to 72 h, respectively (Figure 1C). This delay in appearance enables the artist to exert temporal control over the appearance of different images and patterns within his artwork, as shown in Figure 1D where the central pattern made with agar dots containing 10^5^ conidia in 50 µL YAG agar appeared after 24 h, while a secondary pattern made with agar dots containing 10^3^ conidia in 30 µL YAG agar appeared after 48 h. Furthermore, dot-colonies continued to evolve over the 48 h period, with increased conidia growth gradually changing the color of each dot colony from the initial mycelial-white to the subsequent conidial-brown. Thus, a differential application of dot volumes and inoculant concentrations allowed consistent control over the sequence in which the different visual elements appeared in the painting, and the interplay between their changing hues over time.

### 3.2. Artwork Made by Fungal Dot Painting

Having created the initial proof of the concept in Petri dishes, we tested the techniques scalability into larger designs. We initially created a medium-sized (50 × 70 cm) dot painting on black acrylic glass to heighten the contrast with the whitish mycelial dots (Figure 2). Primary motifs were created using agar dots containing 10^5^ conidia in 50 µL YAG agar, while the background was made using agar dots containing 10^5^ conidia in 30 µL YAG agar. Images taken after incubation for 48 h at 37 °C in an enclosed “wet box” (see Materials and Methods) showed that each agar dot contains mycelium with a unique outline and texture. This enhances the visual interest of the artwork compared to using inanimate material such as acrylic paint to generate the dots.

Finally, three fungal dot painting artworks were prepared on 80 × 120 cm black acrylic glass, using the methods developed above (Figure 3A). An example of an aboriginal art dot painting, which inspired our work, is shown alongside (Figure 3B). The painting is by an aboriginal artist Barbara Weir and is entitled *My Mother’s Country* [20]. Aboriginal paintings serve as a medium to convey one’s heritage and ancestral mythology, usually within the geographic context in which that heritage played out. The present series of three paintings similarly relates to the artist’s heritage, depicting the cities and landscapes in which the artist’s family history unfolded. The use of mycelial culture in this context is essential and integral to the artistic concept. It ties in conceptually to the idea of a landscape or terrain being alive and serving as a cultural catalyst. More importantly, it makes a nuanced statement about cultural development by adding an arbitrariness into the paintings’ development and an inability to ultimately realize them in their pristine imagined form. Time-lapse photography (0–72 h) of a single painting shows how the central motifs of the painting can appear and be subsequently obscured over time (Appendix A). Thus, the developed technique allows the creation of large-scale artworks which unfold and tell their story slowly over time.

### 3.3. Artwork Made by CNC Lathing

We extended the theme of using conidial-infused agar on acrylic glass by generating 3 mm-deep channels and hollows on the surface of 20 mm-thick blocks of acrylic glass and impregnating them with YAG agar containing 10^7^ conidia/mL. The images for these artworks were initially drawn in a digital, vector-file format, based on aerial views of real-world locations from the artist’s family history (Figure 4A,B). The image was then cut into the acrylic glass block using a CNC lathe, creating the required channels. The inoculated YAG agar was then injected into the channels. To prevent desiccation of the YAG agar substrate and distortion of the desired image, corresponding covers of 8 mm acrylic glass were applied, sealing the YAG-infused channels between the two acrylic glass plates. The plates were then connected using adhesive medical tape and inserted into a wooden display frame. The mycelial network developed inside this structure at room temperature over the course of up to 2 weeks (Figure 4C).

This technique allowed us to create a very different visual result from the dot paintings while still leveraging the unique texture of the mycelial and subsequent conidial surface. At the same time, it allowed us to again produce a visually engaging art piece which changes and evolves over time.

## 4. Discussion

The goal of our work was to develop novel forms of fungal art through collaboration between an artist developing the use of non-solid materials and a scientific lab specializing in mycology. In a world of increasing separation between the arts and sciences, and the growing specialization within both disciplines, we believed that the interchange of notions and concepts between us could result in novel art forms.

We created two fungal art forms: (i) fungal dot paintings, where agar droplets infused with fungal conidia are applied and incubated on black acrylic glass, and (ii) fungal etch prints, where agar infused with fungal conidia is applied to etchings on the surface of acrylic blocks.

Although they appear outwardly simple, these art forms were carefully developed with the following concepts in mind: (i) Scalability. The pointillist nature of these drawings, composed of repetitive subunits, allows them to be robustly scaled up in size and complexity. Increased complexity can be achieved by incorporating different species of filamentous fungi, yeast and bacteria, to add diverse colors and textures. Scalability can be adjusted using computer-guided printers to generate the microbial agar droplets onto very large or very small surfaces. Three-dimensional objects and shapes of various sizes and dimensions can be etched and infused with agar inoculated with various microbes. (ii) Temporal dynamism. The art forms we developed are not static. They are composed of living organisms that grow and change over time. Therefore, it is possible to design artworks that change and evolve in countless ways during their exhibition. (iii) Dynamic visual range. The art forms described here can be viewed from a distance so that their constituents aggregate to form an overall image or from nearby, to appreciate shapes, textures and colors, or alternatively, from close-up, in order to perceive the exquisite patterns of the spreading hyphae within each droplet or etching. These can be enhanced by providing suitable backgrounds and lighting to the images. (iv) Controllability. Since the microorganism is contained within a discrete droplet or etching, surrounded by inhospitable plastic surfaces, its ability to spread uncontrollably on the surface of the artwork is curtailed. In addition, since a single concentrated inoculum of one microorganism is applied to each droplet or etching, it can dominate and deplete its limited nutrient source, limiting the likelihood of contamination by other microorganisms. This inoculum effect preserves the clarity and integrity of the work over time, unlike current microbial agar art forms.

In summary, our work opens the door for using mycelial and microbial cultures in novel forms of artistic development. Using relatively simple protocols, engaging and captivating artworks can be created at relatively large scales. These distinctive media provide not only visual interest but also conceptual nuance and depth. We are confident that further exploration of the techniques described here will result in the expression of novel and meaningful artistic statements by others.

## Figures and Tables

**Figure 1 jof-07-01018-f001:**
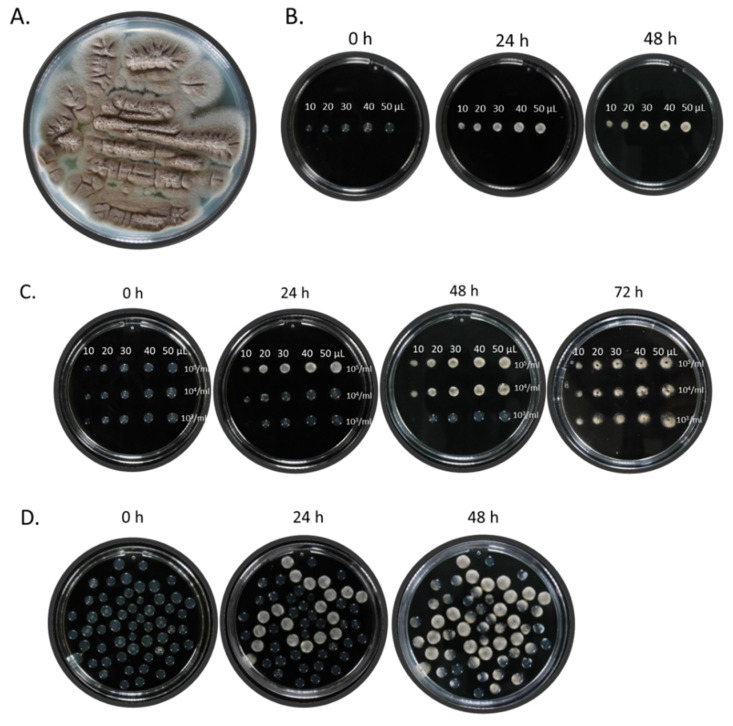
Calibration of the fungal dot painting technique. (**A**) *A. nidulans* strain R153 was grown on YAG agar for 48 h at 37 °C, and conidia were collected and counted. (**B**) 10^5^ conidia/mL were suspended in molten YAG agar at 50–55 °C, point inoculated at 10–50 µL agar volume on the plastic surface of a Petri dish and incubated for up to 48 h at 37 °C. (**C**) 10^3^–10^5^ conidia/mL were suspended in molten YAG agar at 50–55 °C, point inoculated at 10–50 µL agar volume on the plastic surface of a Petri dish and incubated for up to 72 h at 37 °C. (**D**) Conidia were suspended in molten YAG agar at 50–55 °C, point inoculated at either 50 µL agar volume at a concentration of 10^5^ conidia/mL to generate the curved line or at 30 µL agar volume at a concentration of 10^3^ conidia/mL to generate the background.

**Figure 2 jof-07-01018-f002:**
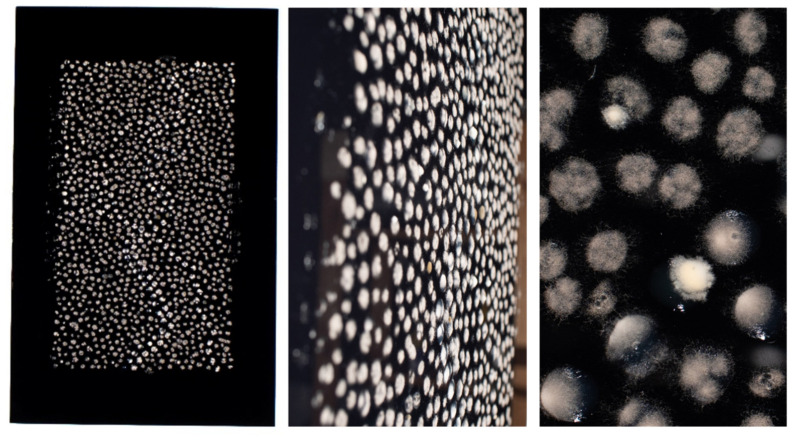
Pilot fungal dot painting on black acrylic glass. A total of 10^5^ conidia/mL were suspended in molten YAG agar at 50–55 °C, point inoculated on black acrylic glass (50 × 70 cm) at 50 µL agar volume to generate the prominent dots or at 30 µL agar volume to generate the weaker background dots. The glass was incubated for 48 h at 37 °C in a humid box. A close-up photograph of the dots (right panel) reveals their heterogeneous mycelial density and texture.

**Figure 3 jof-07-01018-f003:**
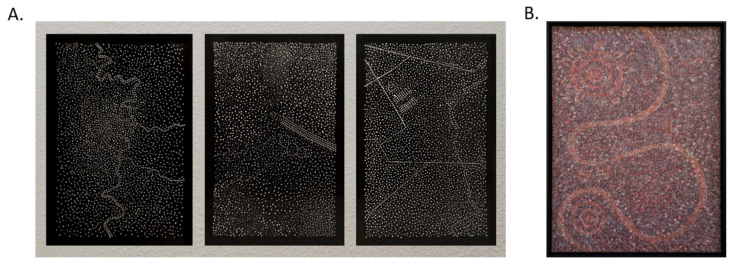
Fungal dot painting artwork. (**A**) A total of 10^7^ conidia/mL were suspended in molten YAG agar at 50–55 °C, point inoculated on three black acrylic glass sheets (120 × 80 cm) at 50 µL agar volume to generate the prominent dots or at 30 µL agar volume to generate the weaker background dots. The glass was incubated for 48 h at 30 °C in a humid box. (**B**) *My Mother’s Country* by Barbara Weir—one of the dot paintings that inspired our work.

**Figure 4 jof-07-01018-f004:**
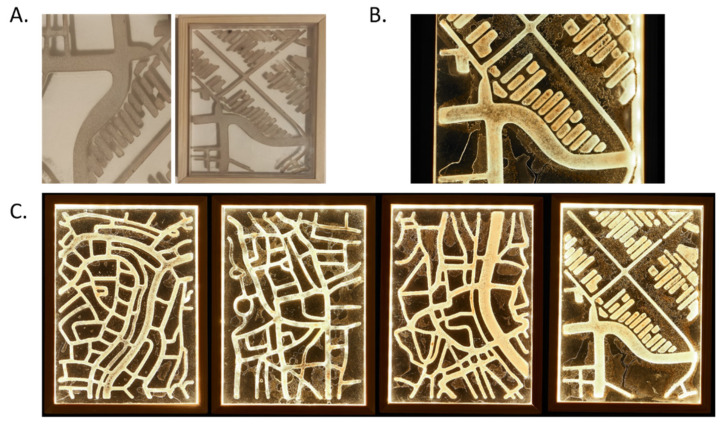
Etched fungal art. (**A**) Schematic showing the CNC lathing of an acrylic block, producing etched channels into which an agar conidial suspension is poured. (**B**) Blocks are incubated at room temperature for up to 2 weeks to allow fungal growth within the channels. (**C**) Selected etched fungal prints in 148 × 210 mm acrylic blocks. Frame and internal lighting were added for display purposes.

## Data Availability

Not applicable.

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
