# Peer review of "Development of Novel Forms of Fungal Art Using Aspergillus nidulans"

_jof, 2021, doi:10.3390/jof7121018_

Round 1

Reviewer 1 Report

The authors described in detail the development of fungal art techniques
using nutrient-rich agar containing A. nidulans conidia spotted on glass acrylic surfaces or impreg-nated onto etched acrylic blocks. This approach generates visually and temporally dynamic art work that is user-friendly, safe, relatively resistant to contamination and easily scalable. Althrough it is interesting, but it not suitable for publication in this scientific journal. It is better to submit to a more special journal.

Author Response

Reviewer 1 requested minor editing which we have performed. Please see the attached manuscript with mark-ups.

Reviewer 2 Report

The manuscript titled "Development of Novel Forms of Fungal Art using Aspergillus nidulans" proposed by Grunwald is quite interesting but irrelevant for most readers of the field of mycology, as implied by the authors themselves in their Introduction.
Although it reads well, only requiring minor edits regarding grammar and other language issues, I believe it won't be appropriate for the readers of JoF. I wish good luck to the authors.

Author Response

Reviewer 2 suggested moderate editing, which we have performed, see the attached manuscript with markups.

Reviewer 3 Report

As the authors themselves note, this is a somewhat unusual submission. I do like it, though, and I enjoyed the ideas and concepts behind it. The text is clear, the concepts well described, and the results (in a sense) speak for themselves.

There are only some very minor clarifications I would like to see;

Fig 3 --(B) I assume that the reproduction is cleared with the journal and artist? (C) is refering to the video, this is currently found in the supp files of the submission. Is this planned as a panel in the final document (and how)? If not, the legend needs correction.

Author Response

Reviewer 3 wrote-

As the authors themselves note, this is a somewhat unusual submission. I do like it, though, and I enjoyed the ideas and concepts behind it. The text is clear, the concepts well described, and the results (in a sense) speak for themselves.

REPLY- We thank the reviewer for these encouraging words.

There are only some very minor clarifications I would like to see;

Fig 3 --(B) I assume that the reproduction is cleared with the journal and artist?

REPLY- Under both US and EU copyright laws, this constitutes permitted fair use and we are legally free to use the image for this article

(C) is refering to the video, this is currently found in the supp files of the submission. Is this planned as a panel in the final document (and how)? If not, the legend needs correction.

REPLY-We would like to see the video embedded in Fig. 3C online